# Application of Prenatal Whole Exome Sequencing for Structural Congenital Anomalies—Experience from a Local Prenatal Diagnostic Laboratory

**DOI:** 10.3390/healthcare10122521

**Published:** 2022-12-13

**Authors:** Theodora Hei Tung Lai, Leung Kuen Sandy Au, Yuen Ting Eunice Lau, Hei Man Lo, Kelvin Yuen Kwong Chan, Ka Wang Cheung, Teresa Wei Ling Ma, Wing Cheong Leung, Choi Wah Kong, Wendy Shu, Po Lam So, Anna Ka Yee Kwong, Christopher Chun Yu Mak, Mianne Lee, Martin Man Chun Chui, Brian Hon Yin Chung, Anita Sik Yau Kan

**Affiliations:** 1Department of Obstetrics and Gynaecology, Queen Mary Hospital, Hong Kong SAR, China; 2Prenatal Diagnostic Laboratory, Tsan Yuk Hospital, Hong Kong SAR, China; 3Department of Obstetrics and Gynaecology, Queen Elizabeth Hospital, Hong Kong SAR, China; 4Department of Obstetrics and Gynaecology, Kwong Wah Hospital, Hong Kong SAR, China; 5Department of Obstetrics and Gynaecology, United Christian Hospital, Hong Kong SAR, China; 6Department of Obstetrics and Gynaecology, Pamela Youde Netherland Hospital, Hong Kong SAR, China; 7Department of Obstetrics and Gynaecology, Tuen Mun Hospital, Hong Kong SAR, China; 8Department of Paediatrics and Adolescent Medicine, The University of Hong Kong, Hong Kong SAR, China

**Keywords:** prenatal whole exome sequencing, prenatal diagnosis, genetic counselling

## Abstract

Fetal structural congenital abnormalities (SCAs) complicate 2–3% of all pregnancies. Whole-exome sequencing (WES) has been increasingly adopted prenatally when karyotyping and chromosomal microarray do not yield a diagnosis. This is a retrospective cohort study of 104 fetuses with SCAs identified on antenatal ultrasound in Hong Kong, where whole exome sequencing is performed. Molecular diagnosis was obtained in 25 of the 104 fetuses (24%). The highest diagnostic rate was found in fetuses with multiple SCAs (29.2%), particularly those with involvement of the cardiac and musculoskeletal systems. Variants of uncertain significance were detected in 8 out of the 104 fetuses (7.7%). Our study shows the utility of WES in the prenatal setting, and the extended use of the technology would be recommended in addition to conventional genetic workup.

## 1. Introduction

Fetal structural anomalies occur in 2–3% of all pregnancies [1]. Mid-trimester anomaly scans are currently offered as part of routine antenatal care in most developed countries. Chromosomal microarray (CMA) is currently offered as a first-line diagnostic test for fetuses with structural congenital anomalies (SCAs) [2], which is useful for identifying copy number variants such as microdeletions or microduplications. While karyotyping can identify an abnormality in 8–10% of cases with SCAs, the addition of CMA could improve the diagnostic yield by 3–10% of cases that might otherwise be missed by conventional karyotyping [3,4,5]. This lack of identification means that most families are left without a specific genetic diagnosis and would have to be counselled based on limited information from the ultrasound findings alone.

The introduction of massively parallel or next-generation sequencing technology, together with the knowledge from human genome mapping, has allowed a widespread application of whole-exome sequencing (WES) in the clinical setting. WES has been found to yield a genetic diagnosis in 25% of children with disorders of probable genetic origin after negative findings in CMA and karyotyping [6,7]. Its high-throughput nature and potential to improve diagnostic yield has also motivated clinicians to consider the use of WES in the prenatal setting. In a recently published systematic review and meta-analysis involving 4350 fetuses for the prenatal WES diagnostic yield of fetal SCAs, WES could provide an additional 31% of diagnosis when CMA/karyotype was non-diagnostic [8]. This is a significant improvement, as accurate genetic diagnosis with realistic counselling is invaluable for parents with fetuses affected by SCAs.

In our locality, the clinical utility of WES in the prenatal setting has yet to be explored. Our study aimed to evaluate the prenatal use of WES in fetuses with SCAs detected on antenatal ultrasound.

## 2. Materials and Methods

### 2.1. Patient and Sample Recruitment

This is a retrospective cohort study of fetuses with SCAs detected on early- to mid-trimester scanning. Ultrasonography was performed by maternal-fetal medicine specialists to confirm the presence of SCAs. Families were recruited at the Maternal Fetal Medicine/Prenatal Diagnostic and Counselling Division in the respective hospitals in Hong Kong. Written informed consent was obtained from all parents after pre-test counselling with verbal and written information. Quantitative fluorescent polymerase chain reaction (QF-PCR) for rapid aneuploidy detection and CMA and/or karyotyping are routinely offered to women undergoing invasive prenatal testing in our unit. WES was provided as a research-based adjunctive option for selected families with suspected monogenetic disease, who received normal QF-PCR and CMA results, after discussion with the clinical geneticist. Families who opt for self-finance WES were also recruited, and samples were sent to a commercial diagnostic laboratory (CENTOGENE, Rostock, Germany). Ethical approval was obtained from the Institutional Review Board (IRB) of the Joint University of Hong Kong—Hospital Authority (Hong Kong West Cluster).

Extraction of genomic DNA was performed from fetal samples, including chorionic villi, amniotic fluid, placental tissues, cord blood, and skin biopsy. Parental peripheral blood samples were collected for trio analysis whenever possible. Maternal cell contamination (MCC) was excluded before further testing. QF-PCR and CMA were performed to rule out common aneuploidies and pathogenic copy number variants before WES.

### 2.2. Trio WES

Trio WES (parents and fetus) was performed from 2017 to 2021 with different WES library kits for validation of clinical utility purposes. The SeqCap EZ Human Exome + UTR Kit (Roche, Penzberg, Germany) and the TruSeq Rapid Exome Library Prep Kit (Illumina Inc., Pleasanton, CA, USA) were used as previously described [9]. The Nextera Exome Enrichment kit (Illumina Inc., Pleasanton, CA, USA) and the Twist Comprehensive Exome (Twist Bioscience, South San Francisco, CA, USA) were used according to the manufacturer’s protocols. Exome libraries were pooled and sequenced using Illumina sequencing platforms, with a target sequencing coverage of 100×. Raw data were analyzed on the in-house bioinformatics pipeline, built according to the Genome Analysis Toolkit (GATK) Best Practices Guideline for germline genetic variations. Sequencing reads were mapped to the reference human genome GRCh37/hg19. Variant call format (vcf) files may also be annotated using Geneyx Analysis (Geneyx Genomex Ltd., Herzliya, Israel), which is a commercial bioinformatics tool for secondary and tertiary NGS data analysis and interpretation [10]. Benign genetic variants were excluded by filtering variants with population frequencies > 0.01. Pathogenic variants were classified according to the American College of Medical Genetics and Genomics (ACMG) guidelines, based on allelic frequency, family segregation, compatibility with phenotypes standardized using Human Phenotype Ontology (HPO) terms, in silico prediction, relevant disease databases, and the literature. Reporting of variants of uncertain significance (VUS), as determined by the ACMG guidelines, was subjected to the decision of a multidisciplinary team consisting of fetal medicine specialists, laboratory scientists, and clinical geneticists.

If patients opted to undergo a self-financed WES for faster turnaround time, their samples were sent out to the commercial laboratory CENTOGENE Company (Centogene GmbH, Rostock, Germany). The DNA library was sequenced on an Illumina platform to obtain at least 20× coverage depth for > 98% of the targeted bases. Data interpretation and analysis were carried out by CENTOGENE. The analysis was performed using an end-to-end in-house bioinformatics pipeline with applications including read alignment to GRCh37/hg19 genome assembly, variant calling, annotation, and comprehensive variant filtering.

Fetal phenotypes were classified into predefined subgroups based on the sonographic findings at the time of the sample collection. Those with more than 1 organ involvement were considered to have multiple SCAs.

## 3. Results

Among the 104 cases recruited in this study, 93 cases underwent an in-house WES, and 11 cases were sent out to a commercial diagnostic laboratory for WES. All SCAs were identified by prenatal ultrasound scan, with 65 (62.5%) fetuses found to have more than one organ system involved. The SCAs were classified into predefined subgroups, including increased nuchal translucency (defined as >3.5 mm)/cystic hygroma (*n* = 25), hydrops (*n* = 7), cardiac (*n* = 38), central nervous system (CNS)-related (*n* = 43), musculoskeletal (*n* = 37), craniofacial dysmorphism (*n* = 17), urogenital abnormalities (*n* = 18), gastrointestinal (*n* = 21), and intrauterine growth restriction (IUGR) (*n* = 7). Ninety-three families were sequenced as complete parental-fetal trios, two families were sequenced as a singleton, three families were sequenced as mother-fetus or father-fetus duo, and six families were sequenced as quadruplets, including a sibling with or without relevant clinical phenotypes. Seventy samples of amniotic fluid, 16 samples of chorionic villi, 15 samples of placental tissues, 1 sample of skin biopsy, and 3 samples of cord blood were obtained from the probands.

For the genetic diagnosis results, diagnostic mutations (pathogenic/likely pathogenic) were detected in 25 fetuses (25/104, 24%) (Table 1). There were 18 pathogenic variants and 10 likely pathogenic variants found in 20 genes. We were also able to identify two fetuses with the Chinese-specific founder mutation, *KLHL40* (Kelch-like family member 40) c.1516A>C, which causes one of the most severe forms of nemaline myopathy (OMIM#615340) [11]. Autosomal dominant diseases were found in 19 fetuses. Two fetuses had presumed de novo heterozygous variants, as their parents received donor gametes, but the donors’ blood were not available for genetic workup. Autosomal recessive diseases were found in six fetuses. VUS was detected in eight fetuses (8/104, 7.7%) (Table 2). There were 11 VUS found in eight genes. Diagnostic genetic variants that were thought to be causative of the SCAs were found in 29.2% (19/65) of those with multiple SCAs, and 15.4% (6/39) of those had single SCAs (Table 3). These variants were pathogenic or likely pathogenic, as classified by the ACMG guidelines. Musculoskeletal and cardiac conditions were the most frequent presentations in fetuses with pathogenic variants identified, while non-immune hydrops fetalis, urogenital anomalies, and IUGR were the least observed.

The pregnancy outcomes were also reviewed. Among the 25 pregnancies found to have diagnostic genetic variants, two resulted in a live birth. Twenty pregnancies ended in termination, two in miscarriage, and one in neonatal death on day 0 of life. Postnatal review, post-mortem babygram, or autopsy results were available in 13 of the cases. The clinical phenotypes and examination results were found to be consistent with the genetic diagnosis. Nine fetuses were found to have maternally and/or paternally inherited diagnostic genetic variants. Their parents were offered the option of preimplantation genetic diagnosis in their future pregnancies to lower the chance of recurrence.

## 4. Discussion

Our study is the largest cohort of prenatal WES published in Hong Kong to date. Through a pilot study of 33 families in Hong Kong [9], we demonstrated the feasibility of prenatal WES. This current study has broadened the inclusion to 104, i.e., 71 new families over the past 4 years. We achieved a diagnostic yield of 24% in identifying causal mutations for the SCAs, which was comparable to other studies [12,13,14], and much improved from our pilot study.

Reaching a genetic diagnosis can provide families with accurate information for genetic counselling and more informed decision-making during the prenatal period. It also allows better anticipation and planning for in-utero interventions, postnatal treatments, symptomatic relief, or palliation. Earlier establishment of genetic diagnosis in the neonatal period has been shown to improve clinical, developmental, and family psychosocial outcomes [15]. It is also found to correlate with shorter neonatal hospital stays and cost savings overall [16]. It would be worthwhile to consider whether prenatal diagnostics could further refine options including antenatal surveillance, delivery, and early postnatal treatment.

The diagnostic yield in our cohort was found to be higher amongst fetuses with multiple SCAs (29.2%) than in those with single SCA (15.4%). This is in line with the findings of previous studies [17]. Biologically, genomic instability is also known to correlate with more complex or severe clinical phenotypes. Specifically, diagnostic genetic variants were most common amongst musculoskeletal and cardiac SCAs, which have also been reported in other cohorts [18]. It was speculated that this was either because single gene variants are more likely to cause anomalies in certain organ systems than others, or that we are simply more experienced in detecting antenatal phenotypes and genotypes involving one organ system than others [19]. We propose that prenatal WES should be considered particularly for fetuses with musculoskeletal and complex cardiac SCAs if their initial genetic workup was negative.

VUS was present in 7.7% of families. The reporting of these variants should be discussed by a multidisciplinary team consisting of laboratory scientists, geneticists, and fetal medicine experts. While the classification and interpretation of variants were based on the knowledge available at the time of reporting, new evidence that has emerged from scientific research and the medical community would certainly aid in re-classification in the future. In one quad analysis in which the ultrasonography of both fetuses (proband and prior pregnancy) showed truncus arteriosus, we identified and reported two compound heterozygous VUS variants on *TMEM260* (Transmembrane protein 260), and the parents were carriers. Subsequently, our team consulted with international expert groups and the case (Family 5) was acknowledged and published as new evidence to expand the mutation spectrum of *TMEM260* biallelic variants in association with structural heart defects and renal anomalies syndrome (SHDRA). The variants were also reclassified as pathogenic/likely pathogenic and may aid future counselling and pregnancy management [20]. Pre-test counselling involving the possibility and implications of VUS is essential in managing the expectations of families undergoing WES. Upon receiving a result of VUS, parents could experience confusion and anxiety [21,22]. The child might also be seen or treated differently by their caregivers during their rearing given the known genetic variation [23]. On the other hand, families may also feel more validated should they consider termination of pregnancy [21]. As WES becomes more widely adopted in the prenatal setting, it would be beneficial to collect anonymous clinical and molecular data onto a common international platform such as ClinVar. This will allow a more confident interpretation of variants and recognition of new genetic syndromes. Another suggested strategy would be to perform targeted exome studies where gene panels are utilized to lower the potential of discovering VUS or other incidental findings [24].

The timing of diagnostic genetic tests is critical for families. Couples often require these results to make decisions regarding termination of pregnancy, continuation of pregnancy, or fetal treatment. Our laboratory’s turnaround time for prenatal WES was, on average, 4 weeks. Couples considering WES were told of the possibility of not receiving the WES results before 24 weeks of gestation, which is the legal time limit to terminate a pregnancy in Hong Kong. For couples considering WES at 20 or 21 weeks, whose WES result might influence their decision of whether to continue with their pregnancy, the option of self-financing to have the sample analyzed at a commercial diagnostic laboratory with a guaranteed turnaround time was offered.

Studies have evaluated the feasibility of offering rapid exome sequencing for ongoing pregnancy with a turnaround time as short as 2 weeks, but the cases were mostly selected with highly specific phenotypes (e.g., skeletal fetal dysplasia), analyzed in trios, and gene panel-based [24,25]. The possibility of shortening turnaround time should be carefully evaluated, given the higher cost and more demanding laboratory manpower. In our cohort, where most families opted for termination of pregnancy, the results of their trios WES were only available upon their post-procedural follow-up. Nonetheless, these results were still invaluable to parents in terms of their future family planning, particularly if pathogenic or likely pathogenic variants had been identified.

All families received detailed post-test counselling by the multidisciplinary team, including fetal medicine specialists, genetic counsellors, and pediatricians. The anticipated clinical features, together with any necessary treatment or precautions for the diagnostic genetic variants, were explained to them. The options of how to manage the pregnancy were discussed. If the WES had been uninformative, the limitations of the test, including the expected coverage, would have been explained. It is crucial for families to understand that a negative prenatal WES does not exclude all genetic causes. The residual risk in such scenarios should be relayed to the family [26]. Periodic re-analysis of cases with a negative WES should be considered. However, until a new genetic diagnosis is revealed, postnatal treatment and review according to the known sonographic features should be planned. Clinicians need to be mindful that the WES results can be stressful, regardless of whether the result is positive or not. Emotional support in the form of counsellors, clinical psychologists, or support groups should be offered throughout their care. The robust pre- and post-test counselling protocol adopted allows families to have good mental preparation surrounding the possible test results and future management.

There are a few limitations to this study. Some of our samples were sent out to another commercial diagnostic laboratory for analysis to achieve a faster turnaround time. The sequencing protocols, laboratory kits, and bioinformatics pipeline for data analysis may differ from our in-house laboratory, potentially affecting the diagnostic yield due to inter-laboratory variations. It is also worth noting that the specifications of the testing platform and the availability of the parental data may affect the diagnostic yield of the study.

## 5. Conclusions

WES is a feasible and promising method for the prenatal diagnosis of fetuses with SCAs and uninformative CMA results. The combination of karyotyping, CMA, and WES will increase the accuracy of molecular diagnosis in the prenatal setting. The improved diagnostic yield allows parents to make informed decisions on both their ongoing and future pregnancies. Nevertheless, the interpretation and counselling of results can be difficult in the presence of incomplete phenotypic data and possible VUS results.

## Figures and Tables

**Table 1 healthcare-10-02521-t001:** List of likely pathogenic and pathogenic disease variants identified by prenatal WES.

Case	Ultrasound Findings	Gestation, Sample Type	Gene	DNA and Protein Alteration, [Inheritance; Zygosity]	ACMG Classification (LP/P)	Clinical Syndrome (OMIM)	Pregnancy Outcome	Postmortem/ Babygram Findings (If Available)
PRE011	Situs inversus, cardiac defects	16 weeks, amniotic fluid	DNAH11	c.13288G>A p.(Gly4430Glu) [Mat] c.8533_8536 delinsATCCG (p.Arg2845Ilefs*31) [Pat] [Compound heterozygous]	LP P	Ciliary dyskinesia, primary, 7, with or without situs inversus (OMIM 611884)	TOP	Situs inversus, fused left and right ventricle, fused adrenal glands
PRE032	Cystic hygroma, dilated left renal pelvis, partial agenesis of corpus callosum, cardiomegaly with right-sided aortic arch	17 weeks, amniotic fluid	RAF1	c.778A>C p.(Trp260Pro) [De novo, heterozygous] (fetus-mother duo WES, Sanger confirmation included paternal blood)	P	Noonan syndrome 5 (OMIM 611553)	TOP	N/A
PRE033	Cystic hygroma, pulmonary atresia—intact ventricular septum	22 + 1 weeks, fetal blood	CHD7	c.2957+1G>A [De novo, heterozygous]	P	CHARGE syndrome (OMIM 214800)	TOP	N/A
HKU-1	Fetal hydrops and cystic hygroma	17 weeks, fetal skin biopsy	PTPN11	c.226G>C p.(Glu76Gln) [De novo, heterozygous]	P	Noonan syndrome 1 (OMIM 163950)	TOP	Cystic hygroma, bilateral pleural effusion
HKU-2	Cystic hygroma, ventricular septal defect, echogenic bowel	17 + 6 weeks, amniotic fluid	PTPN11	c.417G>C p.(Glu139Asp) [De novo, heterozygous]	P	Noonan syndrome 1 (OMIM 163950)	TOP	N/A
HKU-3	Fetal hydrops; atrioventricular septal defect and aortic atresia	21 + 2 weeks, amniotic fluid	CALM1	c.440C>T p.(Thr147Ile) [De novo, heterozygous]	LP	Long QT syndrome (OMIM 616247), Ventricular tachycardia, catecholaminergic polymorphic, 4 (OMIM 614916)	Silent miscarriage at 23 weeks	N/A
HKU-4	Midline cleft lip and palate, hypoplastic left heart, Dandy–Walker syndrome, absent stomach bubble	22 + 3 weeks, amniotic fluid	CHD7	c.2959C>T p.(Arg987*) [De novo, heterozygous]	P	CHARGE syndrome (OMIM 214800)	TOP	Midline clef lip and palate, absent right innominate vein and right innominate artery, aortic isthmic hypoplasia, vermian hypoplasia, right hand 5th digit hypoplasia, unfixed small bowel mesentery
HKU-5	Persistently extended lower limbs, flexed wrists, clenched hands	20 + 4 weeks, amniotic fluid	KLHL40	c.1516A>C p.(Thr506Pro) [Both, homozygous]	P	Nemaline myopathy 8, Autosomal recessive (OMIM 615348)	TOP	Clinical exam showed small ears, flexed hip joints, extended knee joints, flexed elbows. Postmortem exam unremarkable.
HKU-6	Short long bones, bell shaped chest, micrognathia	20 + 6 weeks, amniotic fluid	COL1A1	c.1426G>A p.(Gly476Arg) [De novo, heterozygous]	P	Osteogenesis imperfecta, type II (OMIM 166210)	TOP	N/A
HKU-7	Macrocephaly, bilateral hands, postaxial and bilateral feet, preaxial polydactyly; Paternal history of polydactyly	21 + 4 weeks, amniotic fluid	GLI3	c.710delA p.(Tyr237fs) [Pat, heterozygous]	P	Greig cephalopolysyndactyly syndrome (OMIM 175700)	LB	N/A
HKU-8	Omphalocele, short long bones	14 + 6 weeks, Placental tissue	DYNC2H1	c.1265T>C p.(Leu422Pro) [Mat] c.1540C>T p.(Arg514*) [Pat] [Compound heterozygous]	LP P	Short-rib thoracic dysplasia 3 with or without polydactyly (OMIM 613091)	TOP	Short proximal and distal long bones. All 12 pairs of ribs were short.
HKU-9	Cardiomyopathy with severe mitral and tricuspid regurgitation, hydrops fetalis	19 + 4 weeks, amniotic fluid	SDHA	c.1351C>T p.(Arg451Cys) [Mat; heterozygous]	LP	Cardiomyopathy, dilated, 1GG (OMIM 613642), Mitochondrial complex II deficiency, nuclear type 1 (OMIM 252011)	Preterm LB at 31 weeks, NND on day 0	Dilated cardiomyopathy
HKU-10	Double outlet right ventricle, coarctation of aorta, diaphragmatic hernia	20 + 5 weeks, amniotic fluid	MYRF	c.2014-1G>A [De novo, heterozygous]	P	Cardiac-urogenital syndrome (OMIM 618280)	TOP	Congenital diaphragmatic hernia, double outlet right ventricle with hypoplastic left heart and ventricular septal defect
HKU-11	Rhizomelic shortening of long bones, bowed right femur	20 + 5 weeks, amniotic fluid	COL1A1	c.1714G>A p.(Gly572Ser) [De novo, heterozygous]	P	Osteogenesis imperfecta, type II (OMIM 166210)	TOP	Deformed and short limbs with abnormal bone and fractures
HKU-12	Cystic hygroma, cardiomegaly, short long bones with crossed limbs	12 + 3 weeks, chorionic villi	LZTR1	c.667C>T p.(Pro226Leu) [Pat] c.27dupG p.(Gln10Alafs*24) [Mat] [Compound heterozygous]	LP P	Noonan syndrome 2 (OMIM 605275)	Silent miscarriage at 14 weeks	Cystic hygroma, flexion contracture of elbows, single heart outflow tract
HKU-13	Cleft palate, flat facial profile, micrognathia, short long bones	12 + 5 weeks, chorionic villi	TBL1XR1	c.1211_1217delCAGGGAC p.(Pro404Leufs*10) [De novo, heterozygous]	P	Pierpont syndrome (OMIM 602342)	TOP	Dysmorphic features (prominent forehead, hypertelorism, broad nose, prominent upper lip, large low set ears), increased skin fragility and decrease in elastic fibres, cleft palate, kyphosis with small chest cavity, unfixed small bowel mesentery
HKU-14	Dandy–Walker syndrome, joint contractures with clenched hands and rocker bottom feet, ventricular septal defect	Cord blood	PHGDH	c.488G>A p(Arg163Gln) [Both, homozygous]	LP	Neu-Laxova syndrome 1 (OMIM 256520)	LB	N/A
HKU-15	Acromelia, arthrogryposis, short long bones	16 + 6 weeks, amniotic fluid	COL1A2	c.2133+5G>A Assumed de novo (fetus-father duo)	LP	Osteogenesis imperfecta (type II)	TOP	Absent skull bone, shortened bilateral arms, shortened, and angulated bilateral lower limbs
HKU-16	Short and curved long bones and spine, bilateral talipes, micrognathia	13 + 2 weeks, chorionic villi	FLNB	c.514G>C p.(Gly174Arg) [De novo, heterozygous]	LP	Atelosteogenesis, type I (OMIM 108720)	TOP	Presence of scattered more obvious acellular zones and the presence of multinucleated chondrocytes in the resting cartilage
HKU-17	Short long bones, polydactyly, small chest circumference with pulmonary hypoplasia	22 + 1 weeks, amniotic fluid	PRKACA	c.409G>A p.(Gly137Arg) [De novo, heterozygous]	P	Cardioacrofacial dysplasia 1 (OMIM 619142)	TOP	Short limbs, bilateral post-axial polydactyly, pulmonary hypoplasia with small chest
HKU-18	Thickened nuchal translucency, bilateral talipes, persistently extended knees, flexed wrist and clenched hands	18 + 2 weeks, amniotic fluid	KLHL40	c.1516A>C p.(Thr506Pro) [Both, homozygous]	P	Nemaline myopathy 8, autosomal recessive (OMIM 615348)	TOP	N/A
HKU-19	Short limbs with absent mid-segments, rudimentary left humerus, short femur, undermineralised skull	14 weeks, placental tissue	FLNB	c.5109+1G>T [De novo, heterozygous]	LP	Atelosteogenesis, type I (OMIM 108720)	TOP	N/A
SEND-01	Cystic hygroma, thickened nuchal fold, dolichocephaly, congenital diaphragmatic hernia, dextrocardia, left pleural effusion, oligohydramnios	16 + 6 weeks, amniotic fluid	NR2F2	c.314G>A p.(Arg105His) [De novo, heterozygous]	LP	Congenital heart defects, multiple types, 4 (OMIM 618901)	TOP	N/A
SEND-02	Left-sided congenital diaphragmatic hernia, hypoplastic left heart, ambiguous genitalia	19 weeks, amniotic fluid	PACS1	c.607C>T p.(Arg203Trp) [De novo, heterozygous]	P	Schuurs–Hoeijmakers syndrome (OMIM 615009)	TOP	N/A
SEND-03	Double aortic arch with dominant right aortic arch	16 weeks, amniotic fluid this one with subsequent sequencing the previous affected fetus DNA	CHD7	c.7164+1G>A [Pat, Heterozygous, and estimated 14.3% mosaicism present in sperm and buccal mucosa]	P	CHARGE syndrome (OMIM 214800)	TOP	N/A

**Table 2 healthcare-10-02521-t002:** List of variants with uncertain clinical significance identified by prenatal WES.

Case	Ultrasound Findings	Gestation and Sample Type	Gene	DNA and Protein Alteration	OMIM Phenotypes	Pregnancy Outcome	Postmortem Results (If Available)
PRE003	Ventriculomegaly, small cavum septum pellucidum	22 weeks, amniotic fluid	PACS1	c.2413G>A p.(Ala805Thr) [Mat, heterozygous]	Schuurs–Hoeijmakers syndrome (OMIM 615009)	LB	N/A
PRE004	Cerebellar hypoplasia, mild nuchal edema, both elbows and hip flexed, both knees extended, right club foot, bilateral clenched hands	22 weeks, amniotic fluid	EEF1A2	c.862G>A p.(Glu288Lys) [De novo, heterozygous]	Developmental and epileptic encephalopathy 33 (OMIM 616409), Intellectual developmental disorder, autosomal dominant 38 (OMIM 616393)	TOP	Small chin, posterior cleft palate, rocker bottom feet, abnormal posture with generally thin limbs and decreased muscle bulk, shoulders adducted with flexed elbows, adducted thumbs, clenched overlapping fingers, absent palmer creases, flexed hips with fully extended knees
PRE010	Microphthalmia, hypoplastic optic nerve, agenesis of corpus callosum	18 weeks, placental tissue	DIS3L2	c.410A>G p.(Tyr137Cys), [Mat] c.1826G>A p.(Arg609Gln) [Pat] [Compound heterozygous]	Perlman syndrome (OMIM 267000)	TOP	Agenesis of corpus callosum, hypoplastic optic nerve
PRE013	Agenesis of corpus callosum, ventriculomegaly, cardiac defects	19 weeks, amniotic fluid	LRP2	c.1593C>A p.(Ser531Arg) [Mat] c.10538C>A p.(Ser3513Tyr) [Pat] [Compound heterozygous]	Donnai-Barrow syndrome (OMIM 222448)	TOP	N/A
PRE022	Early onset growth restriction, ventricular septal defect, left-sided diaphragmatic hernia, ambiguous genitalia, ventriculomegaly	22 + 3 weeks, Placental tissue	ATRX	c.1825C>G p.(Pro509Ala) [Undetermined]	Alpha-thalassemia myelodysplasia syndrome, somatic (OMIM 300448), Alpha-thalassemia/mental retardation syndrome (OMIM 301040), Intellectual disability-hypotonic facies syndrome, X-linked (OMIM 309580)	TOP	Ventricular septal defect, bilateral lung hypoplasia, bilateral diaphragmatic hernia, intestinal malrotation
PRE028	Right ventricular hypertrophy, atrioventricular septal defect, persistent left superior vena cava Previous baby had surgically-corrected coarctation of aorta	16 weeks, amniotic fluid	MYH7	c.3803G>A p.(Arg1268His) [De novo, heterozygous]	Dilated or hypertrophic cardiomyopathy (OMIM 613426, OMIM 192600), Laing distal myopathy (OMIM 160500), left ventricular noncompaction (OMIM 613426), Myopathy, myosin storage, autosomal dominant or autosomal recessive (OMIM 608358, OMIM 255160) Scapuloperoneal syndrome, myopathic type (OMIM 181430)	LB	N/A
HKU-20	Truncus arteriosus	20 + 5 weeks, amniotic fluid	TMEM260	c.193-2A>G [Mat] c.1744G>C p.(Glu582Gln) [Pat] [Compound heterozygous] (Remarks: Variants were reclassified and as pathogenic/likely pathogenic. Ref: 20)	Structural heart defects and renal anomalies syndrome (OMIM 617478)	TOP	Truncus arteriosus type 1, single umbilical artery
HKU-13	Cleft palate, flat facial profile, micrognathia, short long bones, small chin, cystic hygroma, increased nuchal translucency	12 + 5 weeks, chorionic villi	PYCR1	c.559G>A p.(Ala187Thr) [Both, homozygous]	Cutis laxa, autosomal recessive, type IIB (OMIM 612940), Cutis laxa, autosomal recessive, type IIIB (OMIM 614438)	TOP	Dysmorphic features (prominent forehead, hypertelorism, broad nose, prominent upper lip, large low set ears), increased skin fragility and decrease in elastic fibres, cleft palate, kyphosis with small chest cavity, unfixed small bowel mesentery

AD: autosomal dominant, AR: autosomal recessive, XLD: X-linked dominant, XLR: X-linked recessive, TOP: termination of pregnancy, N/A: not available; Mat: maternally inherited, Pat: paternally inherited, Both: inherited from both parents; LB: live birth; NND: neonatal death; LP: likely pathogenic; P: Pathogenic. Case numbers starting with PRE were extracted from study cohort in [9]. Case numbers starting with HKU and SEND refer to additional cohorts in 2018–2021.

**Table 3 healthcare-10-02521-t003:** Proportion of diagnosed fetuses (*n* = 25) in each phenotype/SCA category.

Fetal USS	Involved/Total Cases (%)
Single system	6/39 (15.4%)
Multisystem	19/25 (29.2%)
Total	25/104 (24.0%)
**Phenotype Category**	
Thick NT/Cystic hygroma	10/25 (40.0%)
Hydrops fetalis	2/7 (28.6%)
Face/eyes/lip/palate/ears	4/17 (23.5%)
Chest	3/10 (30.0%)
Cardiac	13/38 (34.2%)
Central nervous system	4/43 (9.3%)
Gastrointestinal tract	4/21 (19.0%)
Urogenital	2/18 (11.1%)
Musculoskeletal system	14/37 (37.8%)
Intrauterine growth restriction	2/7 (28.6%)

USS: ultrasound scan, NT: nuchal translucency.

## Data Availability

Data available on request due to restrictions, e.g., privacy or ethical.

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
