# Peer review of "Application of Prenatal Whole Exome Sequencing for Structural Congenital Anomalies—Experience from a Local Prenatal Diagnostic Laboratory"

_healthcare, 2022, doi:10.3390/healthcare10122521_

Round 1

Reviewer 1 Report

Manuscript titled 'Application of prenatal whole exome sequencing for structural congenital anomalies – Experience from a local prenatal diagnostic laboratory' presents retrospective study from respective hospitals in Hong Kong. The aim of this study is to evaluate the prenatal use of WES in fetuses with SCAs detected on antenatal ultrasound on early to mid-trimester scanning.

Manuscript is clear and presented in a well-structured manner and at the appropriate level of English language. It fits to the scope of Healthcare (advanced medical investigation and treatment).

Results are presented as a list of likely pathogenic and pathogenic disease variants identified by prenatal WES. WES is performed after karyotyping, CMA and QF-PCR, if they did not provide valuable explanation for a structural congenital anomalies detected by ultrasound. Manuscript also provides data about pregnancy outcome and postmortem findings.

Although karyotyping, QF-PCR and CMA analysis are available as a standard prenatal techniques offered to women going for an invasive procedure for several years now, in a cases of structural congenital anomalies detected by ultrasound, majority of families will not get specific genetic diagnosis. This study, as well as others, shows importance of offering of WES testing in cases like this.

My only concern is regarding the place and method of WES testing. Since part of the samples were analyzed in another laboratory, there is no data about the method used. So it is not clear how all WES results could be analyzed all together. In my opinion, data from another laboratory, if technical specifications about method used are not available, should be excluded from the study.

Author Response

Thank you very much for your valuable comments.

In cases where a shorter turnaround time is required, especially for parents with pregnancies close to the 24-week legal limit for termination of pregnancy, the option of self-financed WES would be offered. Our clinical samples are sent to Centogene, a commercial laboratory based in Germany ( https://www.centogene.com/), which is well established to perform WES. Their technical methods have been described in detail in clinical reports issued by the company and are comparable to our in-house WES. A brief description of Centogene’s exome analysis has now been included in the revised manuscript. As our study focuses on the clinical application and diagnostic yield of prenatal WES, we felt the place of testing is unlikely to affect the result. We have therefore included these samples within the cohort.

Reviewer 2 Report

This is a paper on WES analysis using prenatal samples in Hong Kong.

Overall, it is a good paper with no major problems.

Some points and concerns

In the "Materials and methods" section, 2.2. Trio WES, there are no details (ver., etc.) of the human reference genome in the 76-98 lines.

Table 1 and 2 are not easy to read. I think it would be better to use symbols for pathogenic and likely pathogenic genes to save space in the vertical columns.

The description of "frameshift" and "nonsense mutation" in Table 1 is not consistent.

In Table 1, it should be indicated whether the variants are previously reported or novel.

In Table 1 and 2, for AD cases in which only heterozygous cases are listed as having a single parental origin, please explain what kind of phenotype the parent had and whether it was diagnosed as a inherited disease or not.

In Table 3, only 25 cases that were diagnosed were evaluated as single system or multi system, but it would be better to state how many of the 104 cases were diagnosed. In line 123, it is stated that 29.2% (19/65) were diagnosed as multi system and 15.4% (6/39) were diagnosed as single system, so it would be better to make this clear at a glance.

In Table 3, it would be better to indicate how many cases were diagnosed in the Phenotype Category out of 104 cases in the same way. This table does not show the total number of cases in each Phenotype Category, so it is difficult to know which Phenotype Category has the highest diagnostic yield.

The Results section, starting from line 111, describes the type of samples, but it says 15 samples of placental tissues and 1 sample of skin biopsy, etc. Is it really a diagnosis by prenatal diagnosis as the title says?

In line 223 of the ''Discussion'', it mentions a commercial diagnostic laboratory instead of an in-house, but there is no detail of the laboratory.
At least for the 11 cases submitted to the commercial diagnostic laboratory, how many laboratories did you contract out to, whether the test was WES or target exome sequencing, whether the platform was illumina or BGI, what library was used, etc. We need to obtain a little more information. I suggest you add the information to the materials and methods section.

Author Response

This is a paper on WES analysis using prenatal samples in Hong Kong.
Overall, it is a good paper with no major problems.

Some points and concerns
In the "Materials and methods" section, 2.2. Trio WES, there are no details (ver., etc.) of the human reference genome in the 76-98 lines.

Reply: Thank you for your comment. We have now included further details of the human reference genome.

Table 1 and 2 are not easy to read. I think it would be better to use symbols for pathogenic and likely pathogenic genes to save space in the vertical columns.

Reply: Thank you for your suggestion. We agree that the tables require further editorial support. We have now used abbreviations and symbols save space in the vertical columns.

The description of "frameshift" and "nonsense mutation" in Table 1 is not consistent.

Reply: We have modified the manuscript accordingly and ensured consistency.

In Table 1, it should be indicated whether the variants are previously reported or novel. Reply: To keep the table as concise as possible, we have not included this information in the table. We felt the table already contains much details  regarding the diagnostic variants. Cases PRE011, PRE032, PRE033 and HKU-18 have previously been reported in our published pilot cohort.

In Table 1 and 2, for AD cases in which only heterozygous cases are listed as having a single parental origin, please explain what kind of phenotype the parent had and whether it was diagnosed as a inherited disease or not.

Reply: For HKU-7, the fetus inherited the GLI3:c.710delA p.(Tyr237Serfs*8) paternally. The father also had polydactyly.

For HKU-9, the fetus inherited SDHA:c.1351C>T p.(Arg451Cys) maternally.  The mother is healthy with no cardiac conditions.

For SEND-03, the father was found to have low level of mosaicism and he is healthy.

For PRE003, the fetus had a maternally inherited PACS1 c.2413G>A p.(Ala805Thr). The mother is apparently healthy with no neurological conditions.

In Table 3, only 25 cases that were diagnosed were evaluated as single system or multi system, but it would be better to state how many of the 104 cases were diagnosed. In line 123, it is stated that 29.2% (19/65) were diagnosed as multisystem and 15.4% (6/39) were diagnosed as single system, so it would be better to make this clear at a glance. In Table 3, it would be better to indicate how many cases were diagnosed in the Phenotype Category out of 104 cases in the same way. This table does not show the total number of cases in each Phenotype Category, so it is difficult to know which Phenotype Category has the highest diagnostic yield.

Reply:

Thank you for your comment. The table has now been revised to include the total number of fetus within each phenotypic category.

The Results section, starting from line 111, describes the type of samples, but it says 15 samples of placental tissues and 1 sample of skin biopsy, etc. Is it really a diagnosis by prenatal diagnosis as the title says?

Reply: All cases within the cohort had prenatal samples obtained via invasive testing methods such as amniocentesis. However, some cases had opted to undergo termination of pregnancy based on the phenotypic features on ultrasonography, without waiting for the genetic results. In such cases, the placental tissue or skin biopsies were also collected in conjunction with the prenatal samples for WES testing to provide fetal diagnosis.

In line 223 of the ''Discussion'', it mentions a commercial diagnostic laboratory instead of an in-house, but there is no detail of the laboratory. At least for the 11 cases submitted to the commercial diagnostic laboratory, how many laboratories did you contract out to, whether the test was WES or target exome sequencing, whether the platform was illumina or BGI, what library was used, etc. We need to obtain a little more information. I suggest you add the information to the materials and methods section.

Reply: Thank you for your comment. Details of the commercial laboratory have now been included under “Materials and Methods” “2.2 Trio WES”.
